# Alpha-Lipoic Acid Treatment Reduces the Levels of Advanced End Glycation Products in Type 2 Diabetes Patients with Neuropathy

**DOI:** 10.3390/biomedicines13020438

**Published:** 2025-02-11

**Authors:** Sára Csiha, Marcell Hernyák, Ágnes Molnár, Hajnalka Lőrincz, Mónika Katkó, György Paragh, Miklós Bodor, Mariann Harangi, Ferenc Sztanek, Eszter Berta

**Affiliations:** 1Division of Endocrinology, Department of Medicine, Faculty of Medicine, University of Debrecen, 4032 Debrecen, Hungary; 2Department of Clinical Basics, Faculty of Pharmacy, University of Debrecen, 4032 Debrecen, Hungary; berta.eszter@med.unideb.hu; 3Doctoral School of Health Sciences, University of Debrecen, 4032 Debrecen, Hungary; 4Division of Metabolism, Department of Medicine, Faculty of Medicine, University of Debrecen, 4032 Debrecen, Hungary

**Keywords:** advanced glycation end products, AGEs, sRAGE, diabetic neuropathy, alpha lipoic acid, atherosclerosis, oxidative stress, ADMA, progranulin

## Abstract

**Background/Objectives:** Type 2 diabetes mellitus (T2DM) and its macro- and microvascular complications are major health concerns with multiple factors, like advanced end glycation products (AGEs), in the background. AGEs induce long-lasting functional modification of the proteins and collagen in the vascular wall and nerve tissue. We investigated the effect of alpha-lipoic acid (ALA) treatment on AGEs, soluble AGE receptor (sRAGE), the AGE/sRAGE ratio, and the parameters of endothelial dysfunction and their correlations. **Methods:** In our 6-month intervention study, 54 T2DM patients with neuropathy treated according to the actual therapeutic guidelines with unchanged oral antidiabetic drugs were included and treated by daily oral administration of 600 mg ALA. A total of 24 gender and age-matched T2DM patients without neuropathy served as controls. **Results:** In our work, we first demonstrated the attenuating effect of alpha lipoic acid therapy on AGEs in humans (11.89 (9.44–12.88) to 10.95 (9.81–12.82) AU/μg (*p* = 0.017)). sRAGE levels or the AGEs/sRAGE ratio were not affected by ALA treatment or by the presence of neuropathy. We found a correlation between the changes of AGEs and the improvement of current perception threshold and progranulin levels, and an inverse correlation with the change of asymmetric dimethylarginine. **Conclusions:** According to our results, ALA decreases AGEs, which may contribute to the clinically well-known beneficial effect in diabetic neuropathy and improvement of endothelial function.

## 1. Introduction

The incidence and the prevalence of type 2 diabetes mellitus (T2DM) are increasing due to sedentary lifestyles and unhealthy eating habits. According to the NCD Risk Factor Collaboration, in 2022, an estimated 828 million adults had diabetes worldwide [1]. Based on the World Health Organization’s opinion, diabetes will be the seventh leading cause of death by 2030 [2,3]. Diabetes mellitus is a complex chronic disease associated with metabolic disturbances and numerous complications. The characteristic hyperglycemia in T2DM is caused by insulin resistance and β-cell damage. Persistent elevated blood glucose contributes to the development of micro- and macrovascular complications [4], which are leading causes of morbidity and mortality [5,6]. Neuropathy, which is estimated to develop in half of the T2DM patients, reduces quality of life and facilitates the development of disability, contributing significantly to healthcare costs [7]. Among its four main types (autonomic neuropathy, proximal neuropathy, distal peripheral neuropathy, and focal neuropathy), distal symmetric polyneuropathy (DSPN) is the most common form [8]. The pathophysiology of diabetes complications is complex. However, it is well established that advanced glycation end products (AGEs) play a significant role in the development and progression of neuropathy, nephropathy, retinopathy, and atherosclerosis. The soluble receptor of advanced glycation end products (sRAGE) is a circulating form of receptor of advanced glycation end products (RAGE) without trans-membrane domain, thereby lacking involvement in the AGEs pathogenesis effect. On the contrary, sRAGE facilitates the degradation and removal of AGEs. The AGE/sRAGE ratio seems to be a promising marker of disease activity in many conditions [9,10].

Alpha lipoic acid (ALA, 1,2-dithiolane-3-pentanoic acid) is a short-chain fatty acid with various pharmacological benefits, such as anti-inflammatory, antidiabetic, antioxidant, anticancer, and neuroprotective effects. Numerous in vitro and in vivo animal studies have shown that ALA treatment can effectively reduce AGE levels, thereby reducing oxidative stress, which ultimately has a beneficial effect on the pathogenesis of neuropathy [11]. ALA is an effective and well-tolerated treatment option in diabetic neuropathy, resulting in a significant improvement of clinical symptoms and signs [12], but the effect of ALA treatment on AGEs has not been investigated in human studies. Preliminary experiments proved that ALA markedly suppressed AGE-induced activation of nuclear factor kappa B (NF-kB) in cultured vascular endothelial cells [13] and retinal endothelial cells [14]. Another study observed that AGE-induced endothelial expression of vascular cell adhesion molecule-1 (VCAM-1) and monocyte attachment to the endothelium were reduced by ALA treatment [15]. Furthermore, ALA prevented the up-regulation of AGE-induced nitric oxide synthase (iNOS) expression and nitric oxide (NO) production in murine microglial cells [16]. ALA can also reduce the AGE-mediated formation of lipid peroxidation products in human neuronal cells [17,18]. In addition, topical ALA nanoparticles significantly reduced RAGE expression and enhanced skin wound healing in streptozotocin-induced damage in diabetic mice [19].

Our research group has previously examined the effect of ALA treatment in T2DM patients with neuropathy on asymmetric dimethylarginine (ADMA) and progranulin (PGRN), and found a decrease of ADMA, which is a known risk factor of atherosclerosis through the inhibition of nitric oxide synthase, and an increase of progranulin, which exerts a role in survival, maintenance, and function of neurons and also protects the vascular endothelium against inflammatory reactions and atherosclerosis [20,21,22]. Reducing AGEs seems to be a promising therapeutical target in various diseases and is under extensive research [23]. In our present study, we aimed to evaluate the assumed beneficial effect of ALA on AGEs in T2DM-caused neuropathy to prove the decremental effect of ALA on AGEs in humans and, therefore, confirm a link between the advanced glycation end pathway in the background and ALA in the treatment of DSPN.

We also evaluated the possible association between AGEs, sRAGE, and the vascular parameters of atherosclerosis, together with the effect of ALA treatment on the investigated parameters. We hypothesized that ALA treatment decreases the serum levels of oxidative stress markers and AGEs. Furthermore, the change in serum AGE levels correlates with the severity of neuropathy characterized by the current perception threshold (CPT).

## 2. Materials and Methods

### 2.1. Study Population

In our present study, we enrolled fifty-four T2DM patients with neuropathy (22 men and 32 women; mean age: 64.15 ± 8.66 years; mean known diabetes duration at the initiation of our study: 12.4 years [interquartile range: 4.1–14.7 years]; duration of diabetic neuropathy: 3.2 ± 1.4 years), and twenty-four gender and age-matched T2DM subjects without neuropathy (duration of diabetes was 12.1 years [interquartile range: 4.0–14.6 years]) as a control group. All participants were recruited from the Diabetic Neuropathy Center of Debrecen, Department of Internal Medicine, Faculty of Medicine, University of Debrecen, Hungary, and provided written informed consent. The study protocol was approved by the Regional and Institutional Ethics Committee, University of Debrecen, Clinical Center (UDCC REC/IEC; 4775-2017) and by the Medical Research Council of Hungary, National Scientific and Ethical Committee (5287-2/2019/EÜIG), and the study was carried out in accordance with the Declaration of Helsinki. This study was not registered in the clinical trial registry, as ALA is widely used in neuropathy treatment. Patients in the intervention group were administered 600 mg ALA (WÖRWAG Pharma GmbH, Böblingen, Germany) orally daily for 6 months. Neither subjects nor controls received treatment for neuropathy before the study. All subjects and controls were treated according to the therapeutic guidelines [24], controlled appropriately with oral antidiabetic agents, and unchanged during the follow-up (metformin, sulfonylurea, and/or dipeptidyl-peptidase-4 inhibitors). In our study, lifestyle counseling was applied due to the actual therapeutic guidelines in both groups. Patients receiving insulin therapy, sodium/glucose cotransporter-2 inhibitors, or glucagon-like peptide-1 receptor agonists were not enrolled. We also excluded patients with a history of diabetic nephropathy (eGFR < 60 mL/min/1.73 m^2^, persistent albuminuria), type 1 diabetes, or diabetic proliferative retinopathy. Patients with known liver disease, autoimmune disease, and hematological and neurological abnormalities that can worsen or cause peripheral neuropathy were excluded. Patients with prior cardiovascular events, myocardial infarction or established coronary artery disease, stroke and peripheral arterial disease, severe congestive heart failure (NYHA class II–IV), subjects with alcoholism or established malignancy, pregnant women, and current smokers were also excluded.

### 2.2. Sample Collection and Routine Laboratory Measurements

Venous blood samples were collected after overnight fasting from the ALA-treated patients before the beginning of ALA treatment and 6 months later, and also from T2DM patients without neuropathy according to the local clinical protocol. Plasma and sera were separated by centrifugation at 2200× *g* for 10 min, aliquoted, and stored in 200 μL aliquots at −80 °C until later measurements.

Routine laboratory analyses (high-density lipoprotein-cholesterol—HDL-C, low-density lipoprotein cholesterol—LDL-C, triglyceride, total cholesterol, glucose, haemoglobin A1c-HbA1c, high sensitivity C-reactive protein (hsCRP), uric acid, and creatinine) were carried out from fresh sera by Cobas c600 autoanalyzer (Roche Diagnostics GmbH, Mannheim, Germany) at the Department of Laboratory Medicine, Faculty of Medicine, University of Debrecen, Debrecen, Hungary. Parameters were determined according to the manufacturer’s recommendation. Non-HDL-C was calculated as the total cholesterol minus HDL-C.

### 2.3. Measurement of Anthropometric Parameters

Body mass index (BMI) was calculated by dividing an adult’s weight in kilograms by their height in meters squared, and abdominal circumference was measured in the horizontal plane midway between the lowest rib and the iliac crest.

### 2.4. Determination of Serum AGE and sRAGE

Serum AGE levels were determined by autofluorescence on a multi-mode microplate reader (Biotek Synegy H1, Agilent Technologies, Santa Clara, CA, USA) and expressed as AU/μg protein as described previously by Münch et al. [25]. The inter-assay coefficient of variation was 8.9% (triplicates on the same day, n = 35). Day-to-day variation (5 days) was 10.5% (n = 35). sRAGE levels were measured by the ELISA method (Human sRAGE Quantikine ELISA kit, R&D Systems Europe Ltd., Abington, UK) according to the manufacturer’s instruction. Sera were used in 3-fold dilution, and sRAGE was expressed as pg/mL. The intra-assay coefficients of variability ranged from 2.6 to 5.3 CV%, and the inter-assay precision ranged from 5.5 to 8.8 CV%. The AGE/sRAGE ratio was also calculated.

### 2.5. PGRN, ADMA, sICAM-1, sVCAM-1, TNFα, oxLDL, and VEGF Measurement

Serum levels of progranulin were measured by a competitive ELISA kit (BioVendor, Brno, Czech Republic) according to the manufacturer’s instructions, with intra-assay coefficients of variability (CVs) ranging from 3.4% to 4.4% and inter-assay CVs ranging from 6.4% to 7.9%. Values were expressed as ng/mL.

Serum ADMA concentrations were determined by the commercially available competitive ELISA method (ADMA-ELISA; DLD Diagnostika GmbH, Hamburg, Germany) according to the instructions of the manufacturer, with intra-assay coefficients of variation (CVs) ranging from 5.7% to 6.4% and inter-assay CVs ranging from 8.3% to 10.3%. The values were expressed as μmol/L.

To determine the serum level of soluble intercellular adhesion molecule-1 (sICAM-1) and sVCAM-1, we used sandwich ELISA kits (Human soluble ICAM-1 and VCAM-1 ELISA, R&D Systems Europe Ltd., Abington, UK). The ELISA method was performed according to the manufacturer’s recommendation. The intra-assay CVs and inter-assay CVs were in the range of 3.7–5.2% and 4.4–6.7% (sICAM-1) and 2.3–3.6% and 5.5–7.8% (sVCAM-1), respectively. The values were expressed as ng/mL.

Serum concentration of tumor necrosis factor alpha (TNFα) was detected using the TNFα ELISA kit (R&D Systems Europe Ltd., Abington, UK) according to the manufacturer’s instructions. The intra-assay CVs ranged from 1.9% to 2.2%, and the inter-assay CVs ranged from 6.2% to 6.7%. Values were expressed as pg/mL.

The oxidized LDL (oxLDL) levels of the serum were measured by the sandwich ELISA method (Mercodia AB, Uppsala, Sweden). The coefficient variations of the intra-assay were from 5.5% to 7.3%, and the inter-assay variations were from 4% to 6.2. The sensitivity was 1 mU/L.

Serum concentration of vascular endothelial growth factor (VEGF) was detected using the ELISA kit (R&D Systems Europe Ltd., Abington, UK) according to the manufacturer’s instructions. The intra-assay CVs ranged from 3.5% to 6.5%, and the inter-assay CVs ranged from 6.7% to 8.5%. Values were expressed as pg/mL.

### 2.6. Assay for Nitrite Concentration

The serum level of nitrite was determined as an indicator of NO production according to the Griess reaction, as described previously. Optical density was detected spectrophotometrically at 550 nm. The concentration of nitrite was determined using sodium nitrite as a standard (10–100 μmol/L). The values were expressed as μmol/L.

### 2.7. Assessment of Autonomic and Peripheral Nerve Function

Peripheral neuropathy was assessed with the application of the Douleur Neuropathique en 4 Questions (DN4) [26] and Neuropathy Total Symptom Score-6 (NTSS-6) [27] questionnaires, vibration perception threshold, and quantitative sensory testing. The exclusion of patients with retinopathy was ensured by an ophthalmologist through in vivo corneal confocal microscopy. We assessed peripheral sensory nerve function using a Neurometer^®^ (Neurotron Inc., Baltimore, MD, USA) and determined CPT, while autonomic function was assessed using Ewing’s five standard cardiovascular reflex tests: changes in heart rate during deep inspiration and expiration heart rate responses to standing up (30/15 ratio), the Valsalva maneuver, systolic blood pressure fluctuation to standing up, and changes in diastolic pressure during sustained handgrip. The severity of autonomic neuropathy was described as the composite autonomic score derived from the results of the tests (normal = 0, borderline = 1, and abnormal = 2), ranging from 0 to 10, with a score of 0–1 classified as normal, 2–3 classified as mild, 4–6 classified as moderate, and 7–10 classified as severe autonomic neuropathy. The applied methodologies have been described in the literature and in our previous work in detail [20,28,29,30].

### 2.8. Statistical Methods

Statistical analysis was performed by STATISTICA^®^ 13.5.0.17 (TIBCO Software Inc., Palo Alto, CA, USA), and graphs were made using GraphPad Prism Version 6.01 (GraphPad software, San Diego, CA, USA). The normality of distribution was checked by the Kolmogorov–Smirnov test, and data were expressed as mean ± standard deviation or median (interquartile ranges). Differences in anthropometric and laboratory parameters between controls and patients with neuropathy prior to ALA-treatment were analyzed by unpaired *t*-test in the case of normal distribution (parametric tests); for non-normal distribution, differences were analyzed with the Mann–Whitney *U*-test (non-parametric tests). In the neuropathy group, prior and after ALA treatment, we determined differences by the paired *t*-test (for data with normal distributions) or the Wilcoxon matched pairs test (for data with non-normal distributions). Sample size calculation for AGEs in neuropathic patients was performed using the Sample Size Calculator of Department of Statistics and Operations Research, University of Vienna (version 1.062). The required sample size for a two-sample Wilcoxon Mann–Whitney *U*-test was minimum n = 8 with 80% (0.8) statistical power. The actual statistical power was 99.97% (0.9997), P(X > Y) = 0.9303, two-sided alpha = 0.05, and n = 54. The Pearson correlation coefficient was used to investigate the connections between variables. The null hypothesis was rejected for two-sided values of *p* < 0.05.

## 3. Results

Table 1 shows the clinical characteristics and laboratory parameters of the patients before and after the ALA treatment, as well as the diabetic control subjects. We did not observe significant differences in glucose, hemoglobin A1c (HbA1C), creatinine, uric acid, hsCRP, lipid parameters, sVCAM-1, sICAM-1, VEGF, and oxLDL regarding patients with and without neuropathy, or in the neuropathy group before and after ALA treatment. We found ADMA and TNFα levels significantly higher in patients with neuropathy compared to controls. CPT and composite autonomic score (CAS) values were also significantly higher in the examined group compared to the control cohort. As a result of 6 months of daily oral ALA treatment, we found that the plasma level of TNFα and ADMA decreased significantly. The levels of NO and PGRN significantly increased. CPT and CAS significantly decreased after ALA treatment.

DN4 is a clinical score suitable to screen and detect the severity of neuropathic pain. A total score of 10 consists of sensory descriptor and examinator elements (description of pain, associated paresthesia/dysesthesia, sensory deficits, evoked pain, sensitivity to touch and pricking, thermal sensory examination, and tactile and pressure allodynia), and the cut-off value for the diagnosis of neuropathic pain is a total score of 4/10 [31]. NTSS-6 also incorporates frequency as a dimension. The scores can vary from 0 to 21.96, and clinically significant symptoms are defined as an NTTS-6 total score of >6 points [27]. We detected significant improvement in the scores from the DN4 questionnaire and NTSS-6 after ALA treatment (Table 1).

Baseline AGE plasma concentration in T2DM patients with neuropathy was 11.89 (9.44–12.88) AU/μg protein, higher than in T2DM controls without neuropathy [9.80 (8.56–10.97) AU/μg protein; *p* = 0.016]. After 6 months of ALA treatment, AGEs significantly decreased to 10.95 (9.81–12.82) AU/μg (*p* = 0.017) (Figure 1A). sRAGE levels did not change significantly in the intervention group with neuropathy during ALA treatment (814.20 (651.70–1094.10) and 806.55 (637.20–1074.6) pg/mL, respectively (*p* = 0.660)). Regarding the sRAGE levels of neuropathic patients compared to the control group’s [685.60 (596.70–978.90) pg/mL], we could not detect a significant difference (*p* = 0.438) (Figure 1B). The AGE/sRAGE ratio was similar in T2DM patients with [2.87 (2.42–3.54) AU/pg] and without neuropathy [2.70 (2.11–3.27) AU/pg]. The AGE/sRAGE ratio did not change significantly during ALA treatment [2.68 (2.12–3.18) AU/pg] (Figure 1C).

Correlations of AGEs, sRAGE levels, and the AGE/sRAGE ratio with vascular parameters of neuropathic patients before and after ALA treatment are shown in Table 2. Regarding AGEs, a negative correlation was observed between AGEs and PGRN before ALA treatment. AGEs correlated with sVCAM-1 and negatively with oxLDL after ALA treatment. We found correlations between VEGF, NO, and sRAGE before ALA, while sVCAM-1, sICAM-1, VEGF, and NO correlated with sRAGE after ALA treatment. Regarding the AGE/sRAGE ratio, we detected negative correlations with sVCAM-1 and VEGF before ALA treatment.

We did not find correlations between AGE and sRAGE and anthropometric and routine laboratory parameters in neuropathic patients before and after ALA treatment. The results are detailed in Appendix A.

The decrease of CPT values and the change of ADMA levels were negatively correlated with the change of AGE levels (Figure 2A,B). The increase of PGRN positively correlated with the change of AGEs (Figure 2C).

## 4. Discussion

### 4.1. Mechanistic Insights

Neuropathy is estimated to appear in half of the patients treated for diabetes in their lifetime, deteriorating their quality of life and increasing morbidity. A long-lasting hyperglycemic state leads to the initiation and propagation of non-enzymatic glycation of nucleic acids, proteins, and lipids in a classical Maillard reaction when reducing sugars react non-enzymatically with an amino group of these macromolecules through a series of reactions forming a Schiff base, followed by an Amadori rearrangement and subsequent oxidative modifications (glycoxidation) to produce AGEs [32]. Indeed, glycemic control is fundamental in the prevention of neuropathy but ineffective in reducing neuropathic pain [33]. Preliminary data suggests that lifestyle intervention resulting in weight loss and increased physical activity might be helpful in managing pain in peripheral neuropathy [34], while lifestyle intervention can also reduce AGEs [35]. Patients with T2DM and components of metabolic syndrome are twice as likely to develop neuropathy compared to patients with T2DM without central obesity, hypertension, insulin resistance, or hyperlipidemia [36], and metabolic syndrome itself is associated with small fiber neuropathy [37].

AGEs have several membrane-bound cell surface receptors, which activate various signal transduction pathways. Some receptors, such as the advanced glycation product receptor 1 (AGER1), help to eliminate AGEs from the body, while others, like receptor of advanced glycation end products (RAGE) induce the activation of the nuclear factor kappa B (NF-κB), mitogen-activated protein kinase (MAPK) and the phosphoinositide-3 kinase (PI-3K) pathways [38], and consequently increase the receptor expression and release of inflammatory cytokines, lead to formation of reactive oxygen species (ROS), activate NADPH oxidase, and induce oxidative stress. The increased oxidative stress itself can also lead to the formation of endogenous AGEs [39], resulting in a self-stimulating process that plays a key role in the appearance of chronic diseases and their complications. AGEs can form cross-links with proteins, thereby altering their function [38,40,41]. Due to cross-links and the irreversible nature of complexes compounding by glucose and tissue protein, AGEs are present for a long time in the tissues of diabetic patients even after glycemic balance is restored, contributing to the so-called “hyperglycemia memory” [10]. These processes induce neuronal damage and apoptosis, thus explaining the role of AGEs in the development of diabetic neuropathy [42].

In the study of Papachristou et al., skin AGEs measured by skin autofluorescence correlated with the presence and severity of DSPN [43]. AGEs might act as early predictors of not only neuropathy, but also macrovascular complications, even in type 1 diabetes [44]. Besides their effect on complications, the accumulation of AGEs also deteriorates insulin sensitivity in adipose tissue and skeletal muscle. The natural route of decrementing AGEs is low ingestion of AGEs, while physical activity acts beneficially through sRAGE increment [45,46,47]. On the other hand, obesity decreases sRAGE, which might contribute to cardiovascular risk [9,48]. Our result regarding similar sRAGE levels in T2DM patients with and without neuropathy is different from earlier data reported in the literature, as Aubert et al. found sRAGE to be positively associated with the presence of peripheral neuropathy [49]. The use of AGEs and the determination of sRAGE generation might serve as potential biomarkers of disease risk and adverse outcomes [50]. Further studies are necessary to clarify these processes.

### 4.2. Clinical Implications

Diabetic neuropathy contributes to healthcare burden cost substantially, not only through medical costs and foot ulcer treatment but also indirectly due to the decline in quality of life and productivity, and even fractures due to neuropathy-caused imbalance [33].

Diabetic neuropathy treatment is largely focused on glycemic control and pain management. After lifestyle alteration, including diet and regular exercise, a significant improvement in the quality of life can be achieved [34]. Regarding the pathogenesis of the disease, it is important to reduce the level of oxidative stress and the formation of AGEs.

Lower ingestion of dietary AGEs, consuming low-processed and antioxidant-rich food, can also have a beneficial effect on the AGE–RAGE axis [51]. In our study, we demonstrated first the AGE-reducing effect of ALA in T2DM patients, which has well-known antioxidant properties.

In a previous study of our group, AGEs were 10.7 AU/μg (mean), while sRAGE was 923 pg/mL (mean) among non-diabetic patients with Hashimoto-thyroiditis, and they were 11.4 AU/μg and 755 pg/mL, respectively, in healthy subjects, which is comparable to our present results of diabetic subjects. Furthermore, in the earlier study, patients treated for hypothyroidism by levothyroxine, which seems to reduce oxidative stress, had lower AGEs than the healthy controls in that study. Furthermore, Quade-Lyssy et al. demonstrated that statins increase sRAGE levels, which might explain the anti-inflammatory effect attributable to statins [52]. In the present study, our patients were taking metformin and statin according to the guidelines, so their AGEs and sRAGE, and thereby their AGEs/sRAGE ratio, might have already been ameliorated by the effect of the abovementioned medicines, similar to the beneficial effect observed among levothyroxine-treated non-diabetic Hashimoto patients [41]. It also seems plausible that the beneficial effect of different pharmacological treatments, and even lifestyle changes like the decrement of dietary AGEs on oxidative stress, might be advantageous in an additive synergetic way. However, the effect of ALA alone on AGEs in the diabetic population is challenging to evaluate, as antihyperglycemic treatment and proper correction of lipid parameters are obligatory early steps of treatment to avoid the development and progression of complications. With enhanced expression of RAGE, low sRAGE levels are described in diabetes, which might cause a change in the redox status in diabetic subjects [53]. In our study, we could not find a difference between subject groups regarding sRAGE.

Concerning vascular parameters, AGE-induced endothelial expression of VCAM-1 and monocyte attachment to the endothelium were reduced by ALA treatment in an in vitro study [15]. VCAM-1 and ICAM-1 mediate inflammation and promote leukocyte migration during inflammation, playing a crucial role in atherosclerosis as they facilitate the occurrence of cardiac events [54]. In our study, sVCAM-1 was similar among patients with and without neuropathy, and we could not detect a change in sVCAM-1 levels before and after ALA treatment. Because of the favorable effect of ALA treatment on adhesion molecule expression in vascular endothelium [55], we expected lower levels of inflammatory markers after the 6-month ALA treatment compared to baseline. Although a significant reduction was observed in this study regarding TNFα levels, the follow-up period is likely too short to observe changes in sVCAM1, sICAM1, and VEGF concentrations. However, there was a correlation between AGE and sVCAM-1 and a moderate non-significant correlation between sRAGE and sVCAM-1 after ALA treatment, while the AGE/sRAGE ratio correlated negatively with sVCAM-1 before ALA treatment in patients with neuropathy. In our study, sVCAM-1 seems to have a stronger correlation with AGEs when the beneficial effects of ALA treatment are present. In our former study, there was a negative correlation between sRAGE and sVCAM-1 described in patients with myocardial infarction [56]. In a recently published work by Sabbatinelli et al., sRAGE was associated with major adverse cardiovascular events in T2DM [57]. AGE–RAGE interaction increases VCAM-1, and GLP-1 inhibits AGE-induced up-regulation of VCAM-1 mRNA levels in endothelial cells by suppressing RAGE expression [58,59]. In our study, we could detect a correlation between sRAGE and sVCAM-1 or between sICAM-1 and VEGF, an inducer of vascular and blood–nerve barrier permeability. In in vitro models, AGEs upregulate and sRAGE downregulate the expression of VEGF [60,61]. In contrast, among our patient groups, we could not detect an association between VEGF and AGEs, but there was a correlation between VEGF and sRAGE, while AGE/sRAGE correlated negatively with VEGF in the neuropathy group before ALA, which seems to strengthen the hypothesis that among T2DM patients, sRAGE might be a risk factor of endothelial dysfunction and consequent vascular events. According to the in vitro findings of Dedert et al., PGRN can help in the clearance of AGEs from cells [62], which is somewhat supported by our findings regarding the correlation between the changes of progranulin and AGEs. With ALA increasing PGRN, the clearance of AGEs is promoted, which results in lower levels of AGEs in 6 months in human subjects, according to our work. It seems plausible that sRAGE might have been higher in the first months of treatment, but this hypothesis needs confirmation.

NO exerts a vasoprotective effect on the vascular wall, and this beneficial effect contributes to the improvement of endothelial function in ACE inhibitor and statin-treated patients [63]. In our study, we found elevated NO levels in neuropathy patients after ALA treatment. We found a correlation between NO and sRAGE in the intervention group before ALA treatment. Besides that, progranulin increased and ADMA decreased after ALA treatment; the beneficial change of both correlated with the change of AGEs. AGEs were found to decrease the expression of the endothelial nitric oxide synthase in human coronary artery endothelial cells in the work of Ren et al. [64], while ADMA is an endogenous inhibitor of nitric oxide production. Progranulin, a new adipokine at the crossroads of metabolic syndrome, diabetes, dyslipidemia, and hypertension, enhances endothelial nitric oxide synthase phosphorylation [65]. All the above-mentioned data support our findings and seem to suggest that ALA exerts a complex effect on nitric oxide synthase through diversified but unidirectional routes.

The detailed effect of AGEs on chronic inflammation and atherosclerosis needs to be further clarified, and the beneficial effect of ALA through neuroprotective and endothelial function improving pathways and their complex relations also need further investigations. Our findings clearly reflect the presence of synergistic neuro- and atheroprotective effects of ALA, in which its AGE-decrementing impact might be fundamental.

### 4.3. Limitations

Some limitations should be mentioned. Despite the appropriate effect size, confirmed by sample size calculation, a larger patient population may improve the statistical power of the study. Our study focused on the Caucasian population; therefore, interpretation of findings for other ethnic groups is only possible to a limited extent. Because of the cross-sectional and prospective nature of our study, any causal relationships should be approached with caution. Moreover, the cause of unchanged sRAGE levels during the ALA treatment is partly unknown. We measured the soluble form of RAGE from human sera, but the exact mechanism of sRAGE generation should be verified during further in vitro studies.

Future research is necessary, as ALA seems to be promising in patients without neuropathy in the prevention of the development of complications in a preclinical state. Our hypothesis should be confirmed on a broader patient group. It would be useful to enroll patients in future research at the diagnosis of T2DM to test the effect of ALA more clearly as a supplementation of dietary changes and physical exercise. To clarify the exact mechanisms of action of ALA on AGEs and endothelium, further in vitro research is needed, where RAGE expression can also be evaluated.

## 5. Conclusions

The precise subcellular effect of AGEs in the development and progression of different neuropathy subtypes needs to be further evaluated and remains to be clarified. However, the fact that ALA improved the perception threshold and composite autonomic score, along with the reduction of AGEs after 6 months of oral ALA treatment without a difference in lifestyle intervention, strongly suggests that ALA alleviates neuropathy signs and symptoms, at least in part, through AGE decrement. The decrease in AGEs also correlated with beneficial changes in ADMA and progranulin, which may indicate the complex clinical effect of ALA, including antiatherogenic, vasculo-, and neuroprotective properties.

## Figures and Tables

**Figure 1 biomedicines-13-00438-f001:**
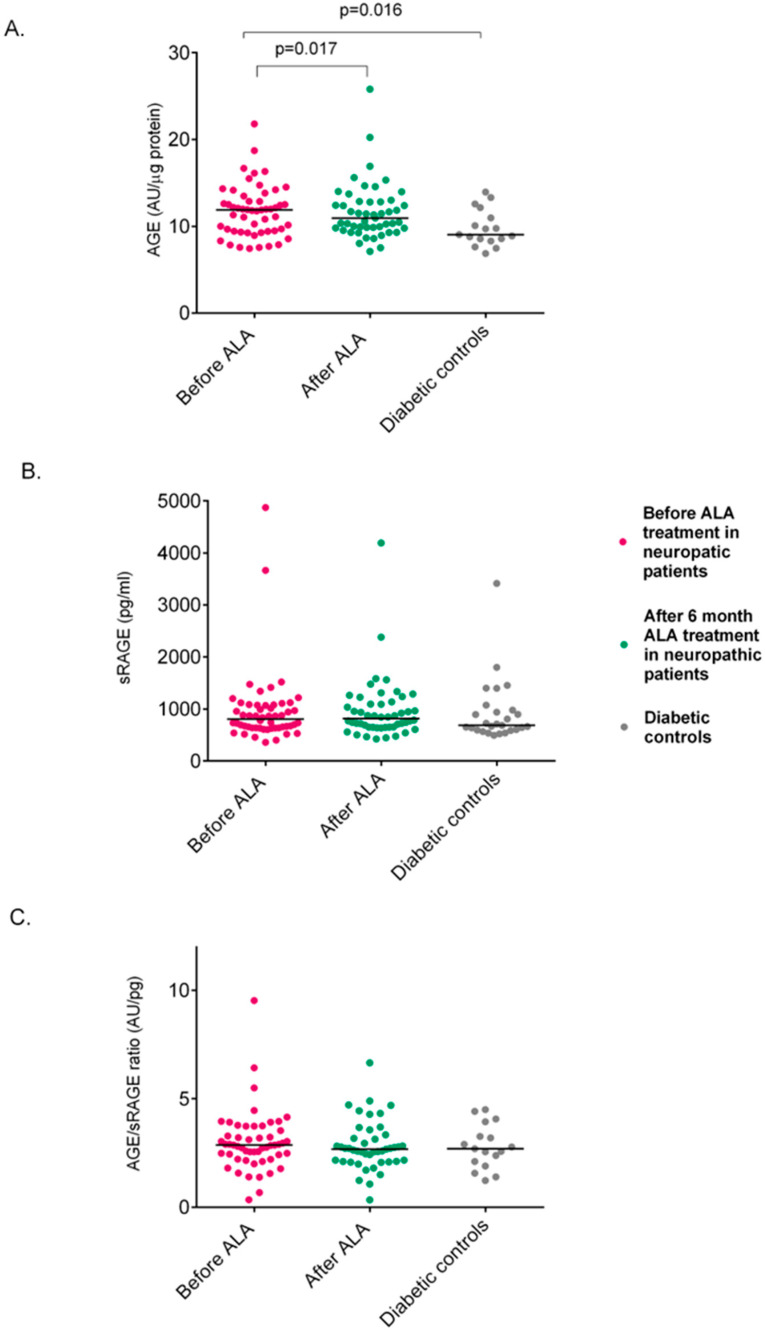
Plasma concentrations of AGEs (**A**), sRAGE (**B**), and AGE/sRAGE ratio (**C**) in patients with type 2 diabetes with neuropathy before and after six months of 600 mg/day alphalipoic acid (ALA) treatment and in diabetic controls without neuropathy. Solid lines represent medians. Magenta dots represent data from neuropathic patients before ALA treatment, green dots represent data from neuropathic patients after six months of ALA treatment, and grey dots represent data of age-, gender-, and BMI-matched diabetic controls. *p*-values were calculated by the Wilcoxon matched paired test in neuropathic patients before and after ALA, and by the Mann–Whitney *U*-test between neuropath patients and diabetic controls. Abbreviations: ALA, alphalipoic acid; AGE, advanced glycation end product; AU, autofluorescence; sRAGE, soluble receptor of advanced glycation end products.

**Figure 2 biomedicines-13-00438-f002:**
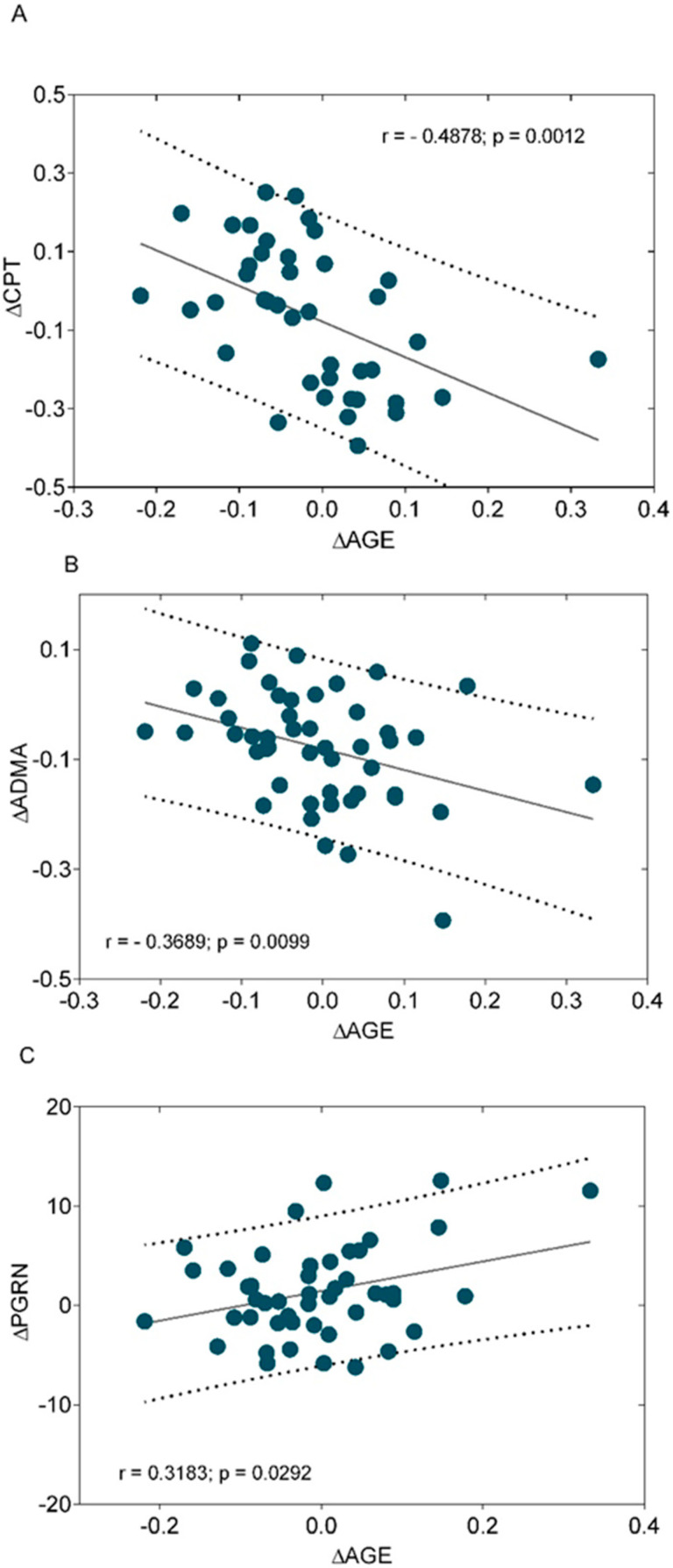
Correlations between the change of current perception threshold (ΔCPT) (**A**), the change of asymmetric dimethyl-arginine (ΔADMA) (**B**), the change of progranulin (ΔPGRN) (**C**), and the change of plasma advanced end glycation product concentration (ΔAGE) after 6 months of alpha-lipoic acid treatment (600 mg daily on the oral route) in patients with type 2 diabetes mellitus with established neuropathy.

**Table 1 biomedicines-13-00438-t001:** Main anthropometric and laboratory parameters of enrolled participants.

	Diabetic Patients with Neuropathy Before ALA	Diabetic Patients with Neuropathy After ALA	Control Diabetic Patients Without Complications
Number of patients (male/female)	54 (22M/32F)		24 (11M/13F)
Age of patients (years)	64.2 ± 8.7		63.6 ± 5.1
Diabetes duration (years)	12.4 ± 2.3		11.3 ± 3.1
Current perception threshold (by Neurometer, mA)	0.473 ± 0.171	0.409 ± 0.154 *	0.375 ± 0.124 **
Composite autonomic score (CAS)	2.67 ± 1.05	1.56 ± 1.24 *	1.13 ± 0.77 **
NTSS-6	8.16 (6.99–15.97)	5.66 (2.99–12.33) *	NA
DN4	3.3 ± 1.4	2.6 ± 1.4 *	NA
BMI (kg/m^2^)	30.02 ± 3.29	29.95 ± 3.73	29.50 ± 2.86
Abdominal circumference (cm)	102.3 ± 12.7	102.4 ± 13.2	101.0 ± 9.8
Glucose (mmol/L)	7.34 ± 2.18	7.51 ± 2.60	7.44 ± 1.36
HbA1C (%)	6.94 ± 0.93	6.84 ± 1.04	6.78 ± 0.75
Creatinine (µmol/L)	72.61 ± 16.97	74.75 ± 14.65	75.17 ± 20.97
Uric acid (µmol/L)	296.51 ± 76.44	304.33 ± 77.69	316.13 ± 57.37
Total cholesterol (mmol/L)	4.84 ± 1.16	4.76 ± 1.24	4.90 ± 1.17
HDL-C (mmol/L)	1.38 ± 0.37	1.38 ± 0.44	1.26 ± 0.33
LDL-C (mmol/L)	2.98 ± 0.97	2.87 ± 1.16	2.84 ± 1.07
Non-HDL-C (mmol/L)	3.47 ± 1.08	3.38 ± 1.19	3.63 ± 1.19
hsCRP (mg/L)	2.1 (0.8–3.36)	2.8 (0.75–5.15)	1.25 (0.9–2.25)
sVCAM-1 (ng/mL)	820 (660–992)	836.6 (674.3–929.6)	729.2 (653.8–847)
sICAM-1 (ng/mL)	210.8 (184.4–247.3)	216.8 (194.4–253.1)	213.3 (189.4–239.4)
VEGF (ng/mL)	62.5 (44.9–93.0)	72.6 (38.6–96.0)	18.6 (15.2–96.0)
Oxidized LDL (U/L)	63.6 (507–91.1)	63.36 (45.59–89.77)	70.76 (59.18–99.46)
TNFα (pg/mL)	1.18 ± 0.36	1.05 ± 0.50 *	0.75 ± 0.29 **
ADMA (µmol/L)	0.61 ± 0.11	0.53 ± 0.11 *	0.56 ± 0.10 **
NO (µmol/L)	16.8 ± 11.1	21.5 ± 9.0 *	19.1 ± 10.9
PGRN (ng/mL)	34.89 ± 7.13	36.23 ± 7.93 *	33.13 ± 7.35

Abbreviations: ADMA, asymmetric dimethylarginine; ALA, alpha-lipoic acid; BMI, body mass index; DN4, Douleur Neuropathique en 4 Questions; HbA1C, Hemoglobin A1c; HDL-C, high-density lipoprotein cholesterol; hsCRP, high-sensitivity C-reactive protein; LDL-C, low-density lipoprotein cholesterol; NA, not available; NO, nitrogen monoxide; NTSS-6, neuropathy total symptom score-6; PGRN, progranulin; sICAM-1, soluble intercellular adhesion molecule-1; sVCAM-1, soluble vascular cell adhesion molecule-1; TNFα, tumor necrosis factor alpha; VEGF, vascular endothelial growth factor. Values are presented as mean ± SD or median (interquartile ranges). * indicates *p* < 0.05 between neuropathic patients before and after ALA treatment (by Student’s paired test or Wilcoxon matched paired test). ** indicates *p* < 0.05 between baseline data of neuropathic patients compared to diabetic controls (by Student’s unpaired test or Mann–Whitney *U*-test).

**Table 2 biomedicines-13-00438-t002:** Correlations of AGE and sRAGE with vascular markers in neuropathic patients before and after ALA treatment.

	**AGE (AU/μg Protein)**			
	**Before ALA**		**After ALA**	
	**r**	** *p* **	**r**	** *p* **
sVCAM-1 (ng/mL)	−0.25	0.068	0.38	0.007
sICAM-1 (ng/mL)	−0.23	0.094	0.21	0.15
VEGF (ng/mL)	−0.21	0.132	−0.02	0.0872
Oxidized LDL (U/L)	−0.14	0.326	−0.28	0.049
TNFα (pg/mL)	−0.10	0.469	0.12	0.4
ADMA (µmol/L)	0.05	0.704	0.23	0.112
NO (µmol/L)	0.02	0.9	0.31	0.093
PGRN (ng/mL)	−0.31	0.022	0.09	0.56
	**sRAGE (pg/mL)**			
	**Before ALA**		**After ALA**	
	**r**	** *p* **	**r**	** *p* **
sVCAM-1 (ng/mL)	0.18	0.196	0.28	0.045
sICAM-1 (ng/mL)	0.20	0.154	0.35	0.013
VEGF (ng/mL)	0.35	0.009	0.32	0.022
Oxidized LDL (U/L)	−0.06	0.66	−0.20	0.155
TNFα (pg/mL)	0.20	0.16	−0.20	0.17
ADMA (µmol/L)	0.24	0.082	0.26	0.063
NO (µmol/L)	0.51	0.002	0.33	0.05
PGRN (ng/mL)	0.16	0.18	0.23	0.104
	**AGE/sRAGE Ratio (AU/pg)**			
	**Before ALA**		**After ALA**	
	**r**	** *p* **	**r**	** *p* **
sVCAM-1 (ng/mL)	−0.29	0.035	−0.08	0.62
sICAM-1 (ng/mL)	−0.25	0.073	−0.13	0.397
VEGF (ng/mL)	−0.38	0.005	−0.28	0.062
Oxidized LDL (U/L)	0.03	0.834	0.08	0.585
TNFα (pg/mL)	−0.17	0.238	−0.16	0.301
ADMA (µmol/L)	−0.21	0.139	−0.06	0.715
NO (µmol/L)	−0.07	0.71	−0.07	0.707
PGRN (ng/mL)	−0.26	0.059	−0.01	0.928

Abbreviations: ADMA, asymmetric dimethylarginine; ALA, alpha-lipoic acid; LDL, low-density lipoprotein; NO, nitrogen monoxide; PGRN, progranulin; sICAM-1, soluble intercellular adhesion molecule-1; sVCAM-1, soluble vascular cell adhesion molecule-1; TNFα, tumor necrosis factor alpha; VEGF, vascular endothelial growth factor. Variables with non-normal distribution were logarithmized before correlation analyses.

## Data Availability

The original contributions presented in this study are included in the article/Appendix A. Further inquiries can be directed to the corresponding author.

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
