# Peer review of "Alpha-Lipoic Acid Treatment Reduces the Levels of Advanced End Glycation Products in Type 2 Diabetes Patients with Neuropathy"

_biomedicines, 2025, doi:10.3390/biomedicines13020438_

Round 1
Reviewer 1 Report
Comments and Suggestions for Authors
The present manuscript is an original research article on a subject of great interest, considering the impact of diabetic neuropathy on the morbidity and mortality of the patients and on the costs involved. The therapeutic options for this condition are not very extensive, so proving additional mechanisms and benefits of the existing drugs is important and it could extend their prescription. The results are consistent and valuable. Please take into consideration the following suggestions for the improvement of the manuscript:
Abstract
- The abstract is well organized, it contains the essential information and it is suitable and in accordance with the whole manuscript
- The second phrase related to AGEs (lines 15-17) should be linked with the first, highlighting the involvement of AGEs in the development of diabetes complications
- The expression ”on stable antihyperglycemic treatment” (line 20) is slightly confusing; please consider to reformulate it
Introduction
- The first statement (lines 35-36) could be sustained by adding some recent epidemiological data
- The statement between lines 42-43 could be developed with some prevalence data and/or with data related to its impact, in order to sustain the importance of the problem
- The expression ”treatment modality” (line 59) should be reformulated (ex. treatment option, therapeutic option etc)
Material and Methods
- The material and methods are described in sufficient detail, the inclusion/ exclusion criteria are clearly mentioned. However, why have you excluded patients on insulin, SGLT 2 inhibitors and GLP receptor agonists?
- The patients were treated for the first time with ALA?
- There were changes in antidiabetic treatment regimen during these 6 months?
- I noticed that BMI was calculated; I consider useful including the abdominal circumference, taking into account that abdominal obesity is associated with insulin resistance, and ALA increases insulin sensitivity
Results and Discussion
- The results are clearly presented and the iconography is suggestive and of good quality
- The composite autonomic score mentioned in Results section, (lines 200-204) should be addressed also in Material and Methods section
- According to the results mentioned in Table 1, the administration of ALA was associated with an increase in the levels of hsCRP, sVCAM1, sICAM1, VEGF. Could you discuss these results?
- The correlations between lines 334-341 should be discussed related to their clinical implications
- A short section related to further research could be included.
Reviewer 2 Report
Comments and Suggestions for Authors
First of all, i want to thanks all authors for their effort in that paper
The introduction provides a good background on T2DM and neuropathy but lacks a clear hypothesis or research question. The authors mention that ALA has been shown to reduce AGEs in animal studies, but they do not clearly state why this study is necessary in humans or what specific gaps in knowledge it aims to fill. Additionally, the rationale for focusing on AGEs and sRAGE is not sufficiently developed. The introduction should clearly articulate the study's objectives and hypotheses.
The exclusion criteria are quite extensive, which may limit the generalizability of the findings. For example, excluding patients with a history of cardiovascular events or those on insulin therapy might exclude a significant portion of the T2DM population. The authors should justify these exclusions and discuss how they might affect the study's applicability.
The control group consists of T2DM patients without neuropathy. However, it is unclear whether these controls were matched for other important variables such as duration of diabetes, glycemic control, or comorbidities. This could introduce confounding factors that are not accounted for in the analysis.
The manuscript states that AGEs were measured by autofluorescence, but it does not provide details on the validation of this method or its sensitivity and specificity. Similarly, the ELISA method for sRAGE is mentioned, but without details on the assay's performance characteristics. These details are essential for assessing the reliability of the results.
The manuscript mentions the use of parametric and non-parametric tests but does not provide a clear rationale for choosing these tests. Additionally, the handling of multiple comparisons is not addressed, which could lead to an increased risk of Type I errors.
Table 1 shows baseline characteristics, but it is unclear whether there were any significant differences between the intervention and control groups at baseline. Any differences should be adjusted for in the analysis to avoid confounding.
The manuscript reports a significant decrease in AGEs after ALA treatment, but the effect size is relatively small (from 11.89 to 10.95 AU/ug). The clinical significance of this change is not discussed. Additionally, the lack of change in sRAGE levels and the AGE/sRAGE ratio is not adequately explained. The authors should discuss whether these findings are consistent with previous studies and what they imply about the mechanism of ALA's action.
The manuscript reports several correlations between AGEs, sRAGE, and vascular markers. However, the interpretation of these correlations is not clear. For example, the negative correlation between AGEs and progranulin before ALA treatment is mentioned, but the biological significance of this finding is not discussed. The authors should provide a more in-depth discussion of these correlations and their potential implications.
The authors conclude that ALA decreases AGEs, which may contribute to its beneficial effects in diabetic neuropathy. However, the discussion does not adequately address the limitations of the study, such as the small effect size of the change in AGEs and the lack of change in sRAGE levels. The authors should discuss whether these findings are clinically meaningful and how they compare to previous studies.
The discussion does not sufficiently address the clinical implications of the findings. The authors should discuss how these results might influence the treatment of diabetic neuropathy and whether ALA could be recommended as a standard treatment based on these findings.
The figure shows the changes in AGEs, sRAGE, and the AGE/sRAGE ratio, but the y-axis labels are not very informative. For example, the y-axis for AGEs is labeled "AU/ug," but it is not clear what "AU" stands for. The authors should provide more detailed labels and legends to make the figures more understandable.
The table provides a lot of information, but it is not always clear what the numbers represent. For example, the values for NTSS-6 and DN4 are given as medians and interquartile ranges, but it is not clear what these scores mean in terms of clinical severity. The authors should provide more context for these values.
The conclusion is somewhat repetitive of the discussion and does not provide new insights. The authors should summarize the key findings and their implications more succinctly and suggest directions for future research.
Minor comments :
· Consider rephrasing the title for conciseness: "Alpha-lipoic acid reduces advanced glycation end products in type 2 diabetes patients with neuropathy."
· In the abstract, clarify the study design (e.g., "a 6-month intervention study") and avoid overstating findings (e.g., "we observed a reduction in AGE levels" instead of "we demonstrated first").
· Fix the typo "inform consent" to "informed consent" in the methods section.
· Specify the timing of blood sample collection relative to ALA treatment in the methods.
· Simplify the explanation of BMI calculation, as the formula is well-known.
· Add a footnote to Table 1 to explain abbreviations (e.g., ADMA, ALA, BMI) for clarity.
· Enhance figure captions by providing more context (e.g., specify the treatment duration and patient group).
· Break up complex sentences in the results section for better readability (e.g., separate findings about AGEs, ADMA, and programulin).
· Organize the discussion into subsections (e.g., "Mechanistic Insights," "Clinical Implications," "Limitations") for improved structure.
· Rephrase awkward sentences in the discussion (e.g., clarify the relationship between AGEs, ADMA, and programulin).
· Ensure all references cited in the text are listed in the references section and remove redundant citations.
· Revise awkward phrasing and grammatical errors throughout the manuscript, favoring active voice where possible.
· Mention whether the study was registered in a clinical trial registry (e.g., ClinicalTrials.gov) in the methods section.
· Ensure consistency in reporting units and abbreviations (e.g., "AU/ug" should be clarified or standardized).
· Provide more context for clinical scores (e.g., NTSS-6, DN4) in Table 1 to help readers interpret their significance.
· Avoid repetitive conclusions; instead, succinctly summarize key findings and suggest future research directions.
Round 2
Reviewer 1 Report
Comments and Suggestions for Authors
The manuscript has been sufficiently improved according to the comments and it is suitable to be published in Biomedicines.
Reviewer 2 Report
Comments and Suggestions for Authors
The authors covered and respond to all my points